# A robust machine learning approach for DC bias prediction in DCO-OFDM based Li-Fi systems

**Marwah Salman**[1,2], **David Siddle**[1], **Yuan Gao**[1]*

1 School of Engineering, University of Leicester, Leicester, United Kingdom, 2 Wasit University, Wasit, Iraq

* yg213@leicester.ac.uk

## Abstract

The direct current (DC) in optical orthogonal frequency division multiplexing (DCO-OFDM) scheme is commonly adopted in light fidelity (Li-Fi) technology as it offers a spectrally efficient solution. A prior study adopted a machine learning (ML)-based solution to predict the optimum DC bias using key parameters, including the statistical properties of the OFDM transmitted signal and a polynomial regression model. However, the model's robustness decreased when the data structure was shuffled, indicating limited generalization to unseen data. This study builds upon that work by utilizing the same dataset and improving the prediction model with advanced ML tools, such as the LazyPredict algorithm (LPA), to systematically evaluate and select a regression model. A robust ML regressor selection process is proposed to ensure the reliability of predictions. Additionally, a comprehensive data analysis is conducted to assess the importance of features affecting the optimum DC bias. The results demonstrate that the ensemble learning algorithm, Random Forest (RF), outperforms other models with an R-squared of 0.953 and an RMSE of 0.233. A Friedman statistical test was applied to validate the results over five iterations of model training. Furthermore, hyperparameter tuning and bootstrap sampling were employed to conduct a deeper investigation into the model's performance and stability. The proposed model significantly enhances the accuracy and robustness of DC bias prediction compared to previous approaches, ensuring consistent performance across different data distributions.

## 1 Introduction

Wireless communications play a vital role in a wide range of industrial and everyday applications. The increasing demand for bandwidth has driven the adoption of orthogonal frequency division multiplexing (OFDM), which is a widely used technique to encode digital data across multiple carrier frequencies. This technique is integral to various technologies, including Wi-Fi, 4G long-term evolution (LTE), radio frequency (RF) communication, and optical wireless communications (OWC). Its key

**Data availability statement:** The data relevant to this paper are from the PLOS ONE article Purnita KS, Mondal MRH (2021) Machine learning for DCO-OFDM based LiFi. PLOS ONE 16(11): e0259955. https://doi.org/10.1371/journal.pone.0259955.

**Funding:** This study was funded by the Higher Committee for Education Development in Iraq (HCED), under the auspices of the Iraqi Prime Minister's Office. The award recipient is Marwah Salman. The University of Leicester has agreed to cover the journal publication fees upon acceptance of the manuscript. The sponsors and funders had no role in the study design, data collection and analysis, decision to publish, or preparation of the manuscript.

**Competing interests:** The authors have declared that no competing interests exist.

advantages include high spectral efficiency, robustness to multipath fading, and scalability [1,2]. To address some expected future issues of Wi-Fi (e.g., limited data transfer rate, lack of security, interference), light fidelity (Li-Fi) has been introduced as a complementary technology to RF communications for indoor and outdoor applications in recent years [3,4]. Compared to traditional RF-based communication systems, Li-Fi offers several benefits, such as high speed, security, and unlicensed bandwidth. Thus, it is becoming a promising technology that complements existing Wi-Fi technologies [5]. OFDM can be used in a different form to achieve the spectral efficiency aspect for the limited bandwidth optical source [i.e., light-emitting diode (LED)]. DC-biased optical OFDM (DCO-OFDM) is a common OFDM variant in optical communications due to its spectrally efficient use compared to other variants such as asymmetrically clipped optical OFDM and Flip OFDM [6].

However, there is a significant challenge hindering the development of DCO-OFDM-based Li-Fi, which is the optimization of DC bias at the transmitter side. If the given DC bias is not sufficient, clipping noise impairments would significantly affect the transmission performance. Conversely, a large DC bias contributes to power inefficiency of the transmission [7]. Therefore, DC bias optimization is formulated as a non-convex optimization problem. Recently, machine learning (ML) has demonstrated broad applicability across diverse domains [8]. In wireless communications, emerging ML-based technologies have offered promising solutions for optimizing optical networks by enabling the network to learn from received signals and optimize its resources [9]. To solve the DC bias optimization problem, an ML-based solution has been explored in the literature to predict the optimum DC bias using the transmitted signal features. For example, reference [10] utilized linear and polynomial regression algorithms to predict the optimal DC bias using a set of signal features and transmission characteristics while satisfying a benchmark bit error rate (BER) constraint. This study showed that the polynomial regression algorithm outperformed the linear algorithm in both R-squared ($R^2$) and root mean square error (RMSE), achieving the highest performance with an $R^2$ of 96.77% and an RMSE of 0.1925.

However, using only these ML regression algorithms could limit the potential application of advanced ML algorithms in DC bias prediction. A specific example is that when the structure of data samples changes, as occurs in data shuffling, the performance of the polynomial algorithm decreases due to its reliance on the learned structural pattern of the given features. This degradation in performance limits the model generalization and thereby affects the prediction of the optimum DC bias on future unseen data.

In addition, the BER feature obtained from the receiver side was included as an input feature, which is infeasible for real-time prediction undertaking at the transmitter side. A detailed literature review is given in the following section, covering conventional methods and ML-based approaches to investigate the performance of DCO-OFDM in general within the Li-Fi context. However, to the best of our knowledge, there are no reported results on the robustness of ML in DC-bias optimization in DCO-ODFM, with the exception of [10], which utilized linear and polynomial models, and this forms the primary motivation of this work.

This paper improves the optimum DC-bias prediction process presented in [10] to ensure efficient and reliable DCO-OFDM transmission. A robust ML regressor selection process using a LazyPredict algorithm (LPA) is proposed to obtain better prediction performance using the same research dataset. An ensemble learning method such as Random Forest (RF) demonstrated superior performance and improved the robustness and generalization of the ML model, making it more applicable to real-world Li-Fi scenarios. Furthermore, the obtained results were subject to statistical validation, such as the Friedman test. The main contributions of this paper are summarized as follows:

1. The problem of DC bias optimization is investigated by exploring a robust ML regressor selection process aided by an advanced regression algorithm called LPA.

2. A comprehensive feature analysis is conducted to evaluate the importance of the features to the optimum DC bias. This process helps in understanding the impact of the relevant features on the model training, ensuring a stable prediction among the transmission cases.

3. We demonstrate that the ensemble learner model (i.e. RF) outperforms the polynomial regression model used in the prior research, indicating an improvement in the prediction as well as the generalization.

4. In the validation, a Friedman statistical test is performed to ensure the robustness and reliability of the developed model performance.

The rest of the paper is organized as follows: Sect 2 gives a summary of the DC bias optimization methods used in literature, including both the conventional methods and the ML based approaches. In Sect 3, brief mathematical fundamentals on DCO-OFDM are provided, along with an overview of the ML benefits in Li-Fi applications. The proposed methodology for ML regressor selection is introduced in Sect 4. In Sect 5, the results are presented and discussed. The conclusion is drawn in last Section.

## 2 Related works

### 2.1 Conventional methods

In these methods, mitigating clipping noise was approached differently. For example, in [11], an adaptive DCO-OFDM scheme was proposed, where large DC bias values were used to mitigate clipping noise during performance evaluation. However, in this study, transmission performance was assessed for various large DC bias levels, assuming a target BER of $10^{-3}$. In contrast, reference [12] independently considered the impact of clipping noise on BER performance, separately from the effects of channel noise. The results showed that this type of impairment significantly affects the performance of DCO-OFDM transmission, particularly for high-order multilevel mapping schemes such as quadrature amplitude modulation (QAM). The study proposed a clipping noise mitigation algorithm and noise cancellation procedure at the receiver to improve BER performance. This improvement came with the requirement for several stages of Fourier transform and maximum likelihood detection at the receiving end, which increased the implementation complexity.

To overcome the impact of clipping noise at the transmitter side, an exhaustive search method was proposed in [13] to determine the optimum DC bias for the transmitted signals. However, this optimization process was achieved under a specific optical power constraint. In [14], a companding technique was proposed to compress the negative peaks of the signal to mitigate the clipping noise for those peaks. An inverse companding process was used at the receiver to recover the compressed negative peaks. The results showed that this method achieved better performance over the conventional DCO-OFDM method and significant improvements, particularly for higher modulation orders. In [15], the mean square error (MSE) between the pre-clipped and clipped DCO-OFDM signals was utilized to determine the optimal DC bias and confine the signal within a given range. The performance improvement was mainly achieved for high modulation orders. In [16], a reduction technique for the peak-to-average power ratio (PAPR) was proposed to mitigate the clipping noise. Thereafter, the optimized DC bias for three types of LED was determined. This method applied a random pilot to rotate the phase of the data sequence and prevent the coherent addition of sub-carriers where high peaks occur. Finally, a classical selection algorithm was utilized to determine the lowest PAPR signal for transmission.

In [17], an adaptive DC bias method, named adaptively biased OFDM (ABO-OFDM), was proposed to optimize the DC bias dynamically based on the negative peaks of the signal. The ABO-OFDM method required parts of the bandwidth to accommodate the modified bias which decreased the spectral efficiency of the transmission. In [18], a low-density parity check coding method (LDPC) was proposed to optimize the DC bias according to the LED power constraints and the clipping noise. Optimized DC bias values were determined for 16, 64, and 256 QAM, respectively. Although this method introduced a considerable increase in complexity, the results showed reduced DC bias levels compared to the conventional DCO-OFDM.

## 2.2 Machine learning methods

ML methods have been explored in the literature to optimize the DC bias in DCO-OFDM systems, aiming to enhance transmission efficiency and mitigate clipping noise distortion. In [10], an ML-based approach was proposed to predict the optimum DC bias using statistical features of the transmitted signal. The study demonstrated superior prediction performance with a polynomial regression compared to a linear model, despite utilizing a relatively small dataset. The optimized DC bias maintained the target BER of the study under an additive white Gaussian noise (AWGN) channel. Beyond optimizing DC bias and mitigating clipping noise at the transmitter, ML-based techniques have also been applied at the receiver. In [19], an artificial neural network (ANN) was employed to mitigate both clipping and channel noise, thus enhancing the overall BER performance. The trained ANN processed the received distorted symbols, reinforcing their correlation with the original transmitted symbols. Experimental results demonstrated a significant BER improvement compared to conventional equalization methods. Similarly, in [20], a deep learning approach using the long short-term memory (LSTM) algorithm was implemented to recover the transmitted symbols at the receiver. The LSTM model achieved performance comparable to the optimal maximum likelihood detection scheme, demonstrating the potential of ML in signal recovery tasks. These findings suggest that ML-driven approaches can substantially enhance Li-Fi transmission. While the integration of ML into optical networks is still evolving, this paradigm shift is increasingly recognized as a promising solution for many optimization issues in network parameters [21].

## 3 DCO-OFDM based ML in Li-Fi system

### 3.1 DCO-OFDM fundamentals and clipping noise

A typical block diagram of the DCO-OFDM transmission scheme is shown in Fig 1. The input data is mapped onto M-QAM symbols, where M is the constellation size used for symbol mapping. The inherent nature of QAM symbols is bipolar and complex, represented in the form: $X = [X_0, X_1, X_2, ..., X_{N-1}]$, where $N$ is the total number of subcarriers. To ensure that the transmitted symbols **X** adhere to Hermitian symmetry and produce a real-valued time domain signal, they must satisfy the following conditions: $X_0 = X_{N/2} = 0$ and $X_l = X_{N-l}^*$ for $0 < l < N/2$, where $X_l$ represents the $l$th data-carrying subcarrier, and $X_l$ and $X_{N-l}^*$ are complex conjugates. The Hermitian requirements reduce the actual data-transmitting subcarriers to $N/2 - 1$. As a result, the complete input vector to the IFFT has the structure:

$$\mathbf{X} = [0, X_1, X_2, ..., X_{N/2-1}, 0, X_{N/2-1}^*, ..., X_2^*, X_1^*] \tag{1}$$

The sequence of symbols **X** is then processed using an IFFT to obtain a discrete-time domain signal $x_m$, which is defined as:

$$x_m = \frac{1}{\sqrt{N}} \sum_{l=0}^{N-1} X_l \exp\left(\frac{j2\pi lm}{N}\right) \tag{2}$$

where $x_m$ signal follows a Gaussian distribution with zero mean and $\sigma^2$ variance for large N, and $m$ denotes the sample index [22].

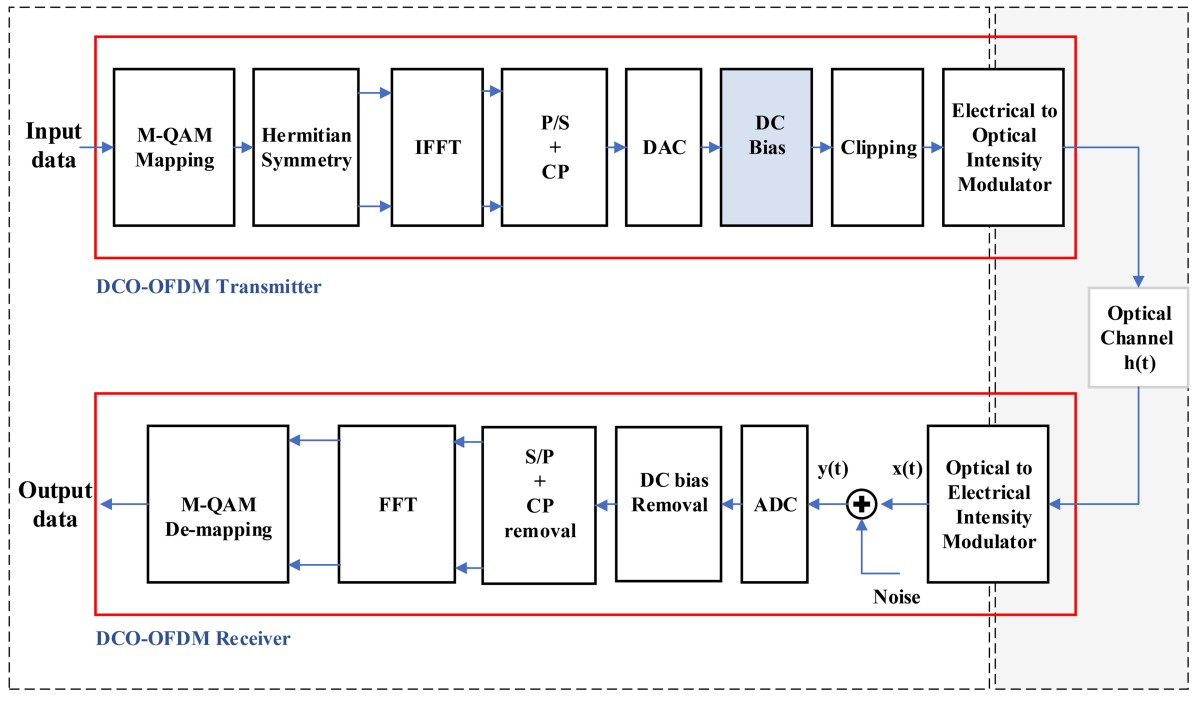

**Fig 1**. Block diagram of DCO-OFDM transmission scheme with QAM modulation.

After undergoing processes such as parallel-to-serial conversion, cyclic prefix (CP) insertion, and digital-to-analog conversion (DAC), the time-domain signal $x_m$ is shifted by an appropriate DC bias and any remaining negative peaks are then clipped to generate the clipped form of the transmitted signal.

The DC bias, denoted as $B_{DC}$, must be positive to ensure a unipolar signal, aligning with the requirements of optical transmission. It is defined as:

$$x_{\text{DCO}}(t) = x(t) + B_{\text{DC}} \tag{3}$$

where

$$B_{\text{DC}} = \mu\sqrt{\mathbb{E}[x^2(t)]} = \mu\sigma_x \tag{4}$$

Here, $\mu$ represents a positive scaling factor for DC bias adjustment, $\sigma_x$ is the standard deviation of the signal, and $\mathbb{E}[\cdot]$ denotes the expectation operator [23]. The average electrical power of the time-domain signal, $P_{\text{elec}}$, is proportional to its variance $\sigma_x^2$. Therefore, the total DC power, $P_{\text{DC}}$, can be expressed as

$$P_{\text{DC}} = |B_{\text{DC}}|^2 + P_{\text{elec}}, \tag{5}$$

$$P_{\text{DC}} = (\mu\sigma_x)^2 + \sigma_x^2 \tag{6}$$

By normalising with respect to $\sigma_x^2$ [24], the expression becomes

$$P_{\text{DC,norm}} = \frac{(\mu\sigma_x)^2 + \sigma_x^2}{\sigma_x^2} = \mu^2 + 1 \tag{7}$$

Finally, the normalised DC power in decibels is given by

$$P_{\text{DC,norm}}(\text{dB}) = 10\log_{10}(\mu^2 + 1) \tag{8}$$

Once the OFDM signals are generated, the optimum DC bias for each signal is determined by iteratively adjusting the scaling factor to ensure the transmission performance remains within a predefined benchmark. A detailed explanation can be found in [10]. The imposed clipping operation to the biased signal causes a clipping noise component that affects the transmission performance. As a result, the DC bias must be accurately calculated to minimize the impact of clipping noise [23,25]. Fig 2 shows the inverse relationship between the clipping noise variance and the DC bias in DCO-OFDM transmission. A larger scaling factor results in a higher DC bias, effectively mitigating clipping noise but at the cost of increased power consumption. Conversely, a smaller scaling factor reduces the DC bias, leading to more pronounced clipping effects, which can introduce nonlinear distortions and degrade the BER performance.

The clipped transmitted signal, $x_{\text{DCO clipped}}(t)$, which accounts for the clipping noise $n_{\text{clipping}}$ is given by:

$$x_{\text{DCO clipped}}(t) = x(t) + B_{\text{DC}} + n_{\text{clipping}} \tag{9}$$

After transmission through the optical medium (i.e. LED), where the typical channel incorporates thermal and shot noise as components of AWGN, $n_{\text{awgn}}$, the received signal $y(t)$ is expressed as:

$$y(t) = x_{\text{DCO clipped}}(t) * h_t + n_{\text{awgn}} \tag{10}$$

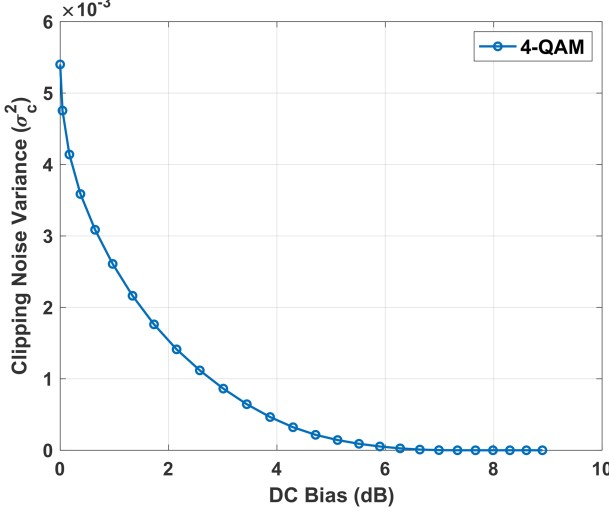

**Fig 2**. The effect of DC bias on clipping noise variance.

Where $h_t$ represents the channel's impulse response. At the receiver, the photodetector converts the received optical power into an electrical signal $y(t)$, which contains both the transmitted signal $x_{\text{DCO clipped}}(t)$ and the noise components introduced by the channel [26,27]. It is crucial to distinguish between clipping noise and channel noise, as they originate from different sources and require separate mitigation strategies. Clipping noise, which results from insufficient DC bias, must be minimized at the transmitter to prevent signal distortion before transmission. In contrast, channel noise, introduced during propagation, can be effectively managed through channel estimation and equalization techniques at the receiver [28].

## 3.2 Machine learning in practical Li-Fi applications

The use of advanced transmission techniques and the implementation of highly flexible principles in optical networks have significantly increased the complexity of their design and operation. This complexity arises from the need to manage various adjustable parameters, such as modulation formats, data rates, and adaptive channel bandwidths. In such flexible use cases, accurately modeling the system with closed-form formulas is often challenging. Recently, ML-based approaches provide the potential to mitigate the nonlinear effects in these networks, which can effectively capture complex behaviors by training on historical network data. The application of ML in physical layer scenarios is primarily driven by the presence of nonlinear effects in optical networks, which render analytical solutions either inaccurate or excessively complex. These nonlinearities can significantly degrade the performance of optical communication systems. However, the practical deployment of ML-based solutions remains challenging due to computational limitations at end-to-end communication terminals, particularly for models that require large volumes of training data [21,29,30].

In the Li-Fi system, the need to provide high-speed connectivity in future applications has opened the door for ML to build intelligent and efficient solutions to overcome many challenges that limit the development phase. So far, many studies have shown that ML plays a crucial role in solving problems such as channel estimation, system optimization, data detection, and decoding [31,32]. In this context, this study aims to develop a robust ML model capable of learning the characteristics of the transmitted signal in a DCO-OFDM system, aiming to enhance system performance by minimizing clipping noise and improving transmission efficiency simultaneously.

## 4 Methodology

In this section, we explain the methods employed to address the research problem. Each subsection provides detailed insights into the specific aspects of our approach.

### 4.1 Data description

In this study, we employed the dataset generated and described by Purnita et al. [10], which was specifically designed for machine learning applications aimed at determining the optimum DC bias in DCO-OFDM systems. The dataset was created using a MATLAB simulation model and consists of 250 samples of DCO-OFDM signals generated under diverse transmission conditions. Each sample captures a combination of system parameters and statistical features that are directly relevant to the DC bias optimization problem. The dataset includes the following key parameters: constellation size of QAM modulation (M), number of subcarriers (N), mean, minimum, maximum, standard deviation of the transmitted signal, optimum DC bias, and the resulting BER. To illustrate the distribution of samples across different (N, M) combinations, Table 1 summarizes the number of samples available for each case. As shown, the dataset is not evenly distributed, with certain (N, M) pairs being more heavily represented than others. The values of N are 256, 512, and 1024, which are commonly adopted in OFDM systems. These values are desirable because they ensure the transmitted signal approaches a Gaussian random variable with approximately 95.6% of signal amplitudes falling within twice the standard deviation of the mean [11].

**Table 1. Distribution of samples across subcarriers (*N*) and modulation orders (*M*).**

| *N*\\*M* | 4 | 16 | 64 | 256 | 1024 |
|---|---|---|---|---|---|
| 256 | 81 | 35 | 23 | 17 | 6 |
| 512 | 14 | 0 | 0 | 13 | 14 |
| 1024 | 0 | 10 | 0 | 21 | 15 |

The modulation order M takes values of 4, 16, 64, 256, and 1024. This wide range represents realistic transmission scenarios by spanning different data rates, with higher orders providing greater throughput but less robustness to clipping noise. The statistical features extracted from each waveform provide a compact representation of the signal and directly inform the DC bias adjustment. Specifically, the minimum and maximum values define the signal's amplitude range, which determines the margin required to avoid clipping distortion. The mean indicates the average offset of the signal, and the standard deviation characterizes the signal's power distribution and variability, which influences the scaling factor that primarily controls the required bias. The dataset also records the corresponding BER, serving as a performance benchmark. Together, these features enable the prediction of the optimum DC bias necessary to maintain transmission reliability.

The simulation model used to generate the dataset considered only an AWGN channel. This choice is justified because AWGN is the dominant noise source in indoor optical wireless communication environments, and the primary objective of DC bias optimization is to mitigate clipping noise at the transmitter before signal propagation. The BER benchmark was therefore selected to ensure clipping noise remained confined to an acceptable level. However, following Table 2 in Purnita et al. [10], which presents feature-importance scores for this dataset, features with minimal contribution to the DC bias prediction were excluded from our analysis.

## 4.2 Machine learning model selection framework

The proposed methodology for developing a robust machine learning (ML) regression model to predict the optimized DC bias in DCO-OFDM systems is illustrated in Fig 3. This process consists of multiple stages, including data pre-processing, model training, performance evaluation, performance validation, and optimal regressor selection, ensuring a systematic and data-driven approach to optimizing DC bias. To build an accurate and reliable ML regression model, a pre-processing stage was conducted to refine the dataset. This step ensures that the data is clean, well-structured, and suitable for model training. Specifically, MinMaxScaler from the scikit-learn Python library was applied to normalize the dataset features, rescaling values between 0 and 1. This normalization enhances model performance by reducing the impact of varying feature magnitudes. Following the normalization phase, the dataset was divided into two subsets: 70% of the data was used for training, and the remaining 30% was used for testing the model. This 7:3 training-to-test splitting ratio ensures that a significant portion of the data is designated for the training phase while reserving a sufficient amount of data for evaluating the model's performance.

Selecting the most effective machine learning regression model for a specific problem application remains a significant challenge for researchers, where several factors can influence a model's performance such as dataset characteristics and model behavior. Therefore, a comprehensive analysis is crucial to evaluate the model's capability and effectiveness [33]. In this study, a comprehensive evaluation of multiple ML regressors was performed using the LPA to identify the most effective ML regression model. LPA is an advanced Python library designed to automate the comparison of various ML models. The implementation of the LPA library is not yet widely explored but it enables the evaluation of a pool of different regressors [34,35]. This approach is particularly useful for initial regression model selection tasks because it allows us to compare the performance of 41 regressors across various metrics [36]. Table 2 summarizes the learning methods and their corresponding ML regressor in LPA. To mitigate potential biases, the LPA model was trained iteratively five times,

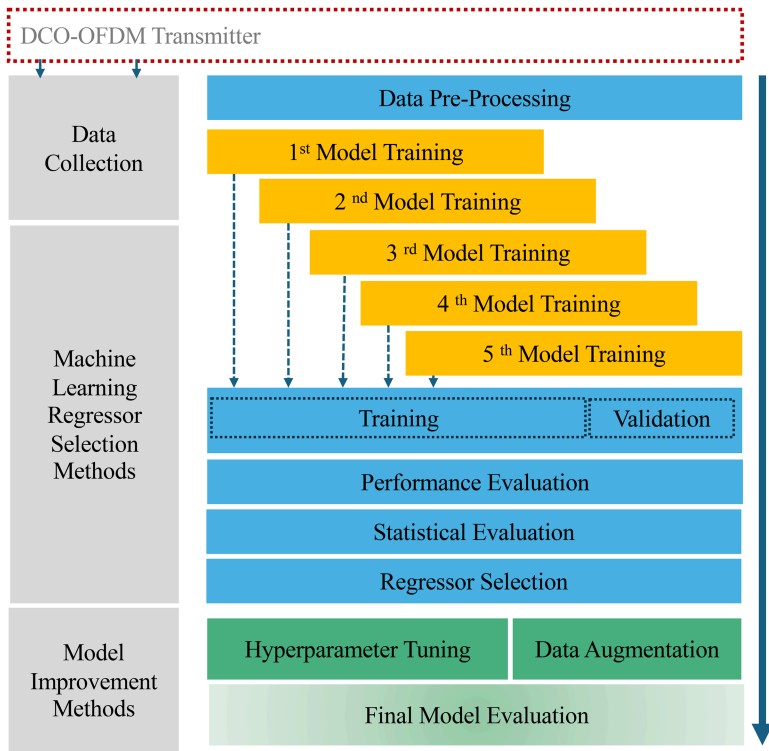

**Fig 3**. **Overview of the proposed methodology for the ML regressor selection process and model evaluation.**

with the dataset being randomly reshuffled before each iteration. This iterative process prevented the model from learning any underlying patterns specific to the particular data collection during the simulation. At each iteration, model performance was evaluated using key metrics consistent with those in [10], and the mean performance across all five iterations was calculated to determine overall effectiveness. To validate the statistical significance of the results and confirm that performance differences were not due to chance, a Friedman statistical test was applied across all iterations. Finally, the most effective ML regression model was selected based on its predictive accuracy and statistical validation, ensuring a reliable and data-driven approach to optimizing DC bias in DCO-OFDM systems.

## 4.3 Machine learning model evaluation

To evaluate the performance of each regression model, the coefficient of determination ($R^2$ or R-squared) and RMSE are used primarily to facilitate comparison with related research. The R-squared measures the proportion of variance in the target variable, giving the best prediction at a value close to or equal to 1. Meanwhile, RMSE measures the root mean of the squared differences between the actual and predicted values, reaching 0 for the most accurate prediction. The mathematical representation of $R^2$ and RMSE are shown below:

$$R^2 = 1 - \frac{\sum_{i=1}^{m}(X_i - Y_i)^2}{\sum_{i=1}^{m}(\overline{Y} - Y_i)^2} \tag{11}$$

**Table 2. ML regression methods and corresponding regressors used in LPA.**

| Regression Methods | Regressors |
|---|---|
| Tree-based and Ensemble Methods | Decision Tree |
| | Extra Tree |
| | Random Forest |
| | Gradient Boosting (XGBoost, LightGBM) |
| | Hist-Gradient Boosting |
| | Bagging |
| | AdaBoost |
| Support Vector Machines (SVM) | SVR |
| | Nu SVR |
| | Linear SVR |
| Linear Regression and Variants | Ridge |
| | Ridge CV |
| | Lasso |
| | Lasso CV |
| | Elastic Net |
| | Elastic Net CV |
| Nearest Neighbors | KNeighbors |
| Neural Networks | Multilayer Perceptron (MLP) |
| Generalized Linear Models | Gamma |
| | Poisson |
| | Tweedie |
| Bayesian Methods | Bayesian Ridge |
| Kernel Methods | Kernel Ridge |
| Specialized Regressors | Dummy Gaussian Process |
| Other Linear Models | Lars |
| | Orthogonal Matching Pursuit |
| | RANSAC |
| | Stochastic Gradient Descent (SGD) |
| | Passive Aggressive |
| | Transformed Target |

$$RMSE = \sqrt{\frac{1}{m}\sum_{i=1}^{m}(X_i - Y_i)^2} \tag{12}$$

where $X_i$ and $Y_i$ are the $i$th predicted and actual values, respectively, while $\overline{Y}$ represents the mean of the actual values.

The Friedman statistical test was conducted in this analysis under the assumption of a non-parametric data distribution, making it particularly suitable for small datasets such as the one used in this study. This test compares the differences in performance scores among LPA regressors over multiple iterations. This procedure ensures that the evaluation of our model is not influenced by random variability. If a p-value of less than 0.05 is achieved during the analysis, this indicates that the observed differences in model performance are statistically significant [37].

The Friedman test is widely regarded as a robust method for evaluating multiple models or algorithms in regression and classification studies, particularly when assumptions of normality or homogeneity of variance are not satisfied [38]. When these assumptions aren't met, the Friedman test is used instead of the traditional statistical tests such as analysis of variance (ANOVA) because it does not rely on strict assumptions [39]. These considerations reinforce the reliability and generalizability of the findings, even with the relatively small dataset. The performance scores were obtained from 5 iterations of running the model while ensuring data reshuffling with each iteration. The mean values of these scores were utilized to identify the most suitable ML regressor for the dataset.

## 4.4 Final regression model improvement

After selecting the ML regressor, a hyperparameter tuning procedure was conducted using a grid search method to optimize the parameters of the selected ML model and improve the prediction performance. In this method, an exhaustive search is used where a pre-defined set of hyperparameters is specified in advance, and the ML model is trained and evaluated for each combination of these parameters. Considering the relatively small dataset size employed in this study, a data augmentation method, known as Bootstrap sampling, was investigated to increase the dataset's size and evaluate the model performance with larger training samples.

## 5 Results and discussion

In this section, we present the key findings of our research and explicit comparison with the previous research outcomes.

### 5.1 Model training and regression model selection

By leveraging our proposed methodology, the model performance was evaluated based on the $R^2$ and RMSE metrics, as shown in Table 3 and Table 4, respectively. Notably, the best performing regressors identified in these tables consistently achieved $R^2$ in the range of 0.8 to 0.9, demonstrating the robustness of our approach across these regression models. To address robustness and generalization concerns regarding the results of the proposed model, particularly given the relatively small size of the dataset, a rigorous statistical evaluation was conducted using the Friedman test. This evaluation compares the performance of the regressors across five iterations of the model training process. Fig 4 presents the Friedman statistics (right axis) alongside the corresponding p-values (left axis) for each iteration. From the second iteration, the Friedman test yields p-values of 0.01, indicating statistically significant performance differences among the regressors. To complement the p-values, the effect sizes, expressed as epsilon-squared ($\varepsilon^2$), ranged from 0.90 to 1.00, reflecting a very strong proportion of variance explained by these differences. Taken together, these results demonstrate that the regressors differ in a statistically significant manner, and that the observed differences are also substantial and practically meaningful.

These findings provide strong evidence of the model's robustness, demonstrating that its performance scores are unlikely to have resulted from random variability [40]. To evaluate the findings, only the mean values of the performance metrics were considered to select the ML regression model. The RF regression model is identified as the best performing model, achieving $R^2$ of 0.95384 and RMSE of 0.2339. This performance demonstrates the model's robustness in predicting the optimized DC bias over the remaining regressors, as visualized in Figs 5 and 6. Given this substantial performance, the model is well suited for practical integration into Li-Fi systems. Its implementation can be effectively aligned

**Table 3. Performance evaluation of ML regression models using $R^2$.**

| Model | $R_1^2$ | $R_2^2$ | $R_3^2$ | $R_4^2$ | $R_5^2$ | $R^2(Mean)$ |
|---|---|---|---|---|---|---|
| **Random Forest** | 0.9657 | 0.9489 | 0.9461 | 0.9437 | 0.9648 | **0.9538** |
| Extra Trees | 0.9463 | 0.9526 | 0.9562 | 0.9467 | 0.9657 | 0.9535 |
| Ada Boost | 0.9541 | 0.9542 | 0.9488 | 0.9481 | 0.9586 | 0.9527 |
| Gradient Boosting | 0.9562 | 0.9449 | 0.9415 | 0.9417 | 0.9621 | 0.9492 |
| Bagging | 0.9566 | 0.9561 | 0.9340 | 0.9443 | 0.9551 | 0.9492 |
| LGBM | 0.9491 | 0.9492 | 0.9573 | 0.9359 | 0.9533 | 0.9489 |
| Hist-Gradient Boosting | 0.9488 | 0.9466 | 0.9547 | 0.9366 | 0.9535 | 0.9480 |
| XGB | 0.9451 | 0.9381 | 0.9274 | 0.9318 | 0.9637 | 0.9412 |
| KNeighbors | 0.8818 | 0.9569 | 0.9528 | 0.8885 | 0.9599 | 0.9279 |
| Extra Tree | 0.9086 | 0.9231 | 0.9316 | 0.9388 | 0.9309 | 0.9266 |
| Decision Tree | 0.9321 | 0.9212 | 0.8869 | 0.9397 | 0.9448 | 0.9249 |
| NuSVR | 0.8864 | 0.9441 | 0.9418 | 0.8769 | 0.9230 | 0.9144 |
| SVR | 0.8898 | 0.9402 | 0.9419 | 0.8738 | 0.9161 | 0.9123 |

**Table 4. Performance evaluation of ML regression models using RMSE.**

| Model | $RMSE_1$ | $RMSE_2$ | $RMSE_3$ | $RMSE_4$ | $RMSE_5$ | $RMSE(Mean)$ |
|---|---|---|---|---|---|---|
| **Random Forest** | 0.1981 | 0.2504 | 0.2548 | 0.2580 | 0.2085 | **0.2339** |
| Extra Trees | 0.2477 | 0.2410 | 0.2295 | 0.2508 | 0.2058 | 0.2349 |
| Ada Boost | 0.2290 | 0.2370 | 0.2482 | 0.2476 | 0.2261 | 0.2375 |
| Gradient Boosting | 0.2239 | 0.2600 | 0.2654 | 0.2626 | 0.2165 | 0.2456 |
| Bagging | 0.2227 | 0.2320 | 0.2818 | 0.2566 | 0.2357 | 0.2457 |
| LGBM | 0.2421 | 0.2497 | 0.2267 | 0.2752 | 0.2403 | 0.2466 |
| Hist-Gradient Boosting | 0.2421 | 0.2560 | 0.2336 | 0.2737 | 0.2398 | 0.2490 |
| XGB | 0.2506 | 0.2756 | 0.2957 | 0.2838 | 0.2119 | 0.2635 |
| KNeighbors | 0.3676 | 0.2298 | 0.2384 | 0.3630 | 0.2227 | 0.2843 |
| Extra Tree | 0.3233 | 0.3071 | 0.2869 | 0.2690 | 0.2923 | 0.2957 |
| Decision Tree | 0.2786 | 0.3108 | 0.3690 | 0.2668 | 0.2618 | 0.2973 |
| NuSVR | 0.3604 | 0.2618 | 0.2646 | 0.3814 | 0.3985 | 0.3153 |
| SVR | 0.3550 | 0.2708 | 0.2644 | 0.3861 | 0.3221 | 0.3196 |

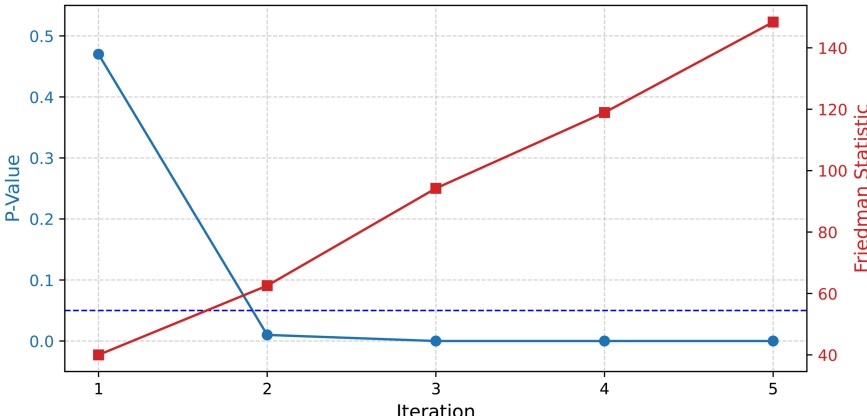

**Fig 4**. **Friedman statistic and p-value across iterations.**

with a block-based approach for adaptive transmission, where the DC bias is optimized for each individual OFDM block. This strategy reduces the frequency of bias updates, improving computational efficiency while maintaining reliable transmission performance [36]. Recent advances, such as early stopping mechanisms [41,42], demonstrate strategies for reducing energy consumption in microcontrollers, which is particularly relevant for future implementations of ML-based Li-Fi systems on embedded hardware. Similarly, data-parallel RF approaches [43,44] highlight opportunities to accelerate training times. While training is performed offline in the current work, such methods could facilitate periodic model updates in dynamic environments, further supporting the long-term viability of ML-driven parameter optimization in Li-Fi networks.

## 5.2 Computational complexity

In our study, all simulations and model training were performed on Apple MacBook M2 Pro chip (10-core CPU, 16-core GPU, 32 GB RAM). It is important to emphasize that the primary computational burden lies in the offline training phase; once the model is deployed, only inference is required in real-time operation. The selected RF regressor is computationally lightweight at the inference stage, as it only requires the system to provide the key parameters for predicting the optimum DC bias. When the model was tested for a block-based approach, the average inference time was approximately

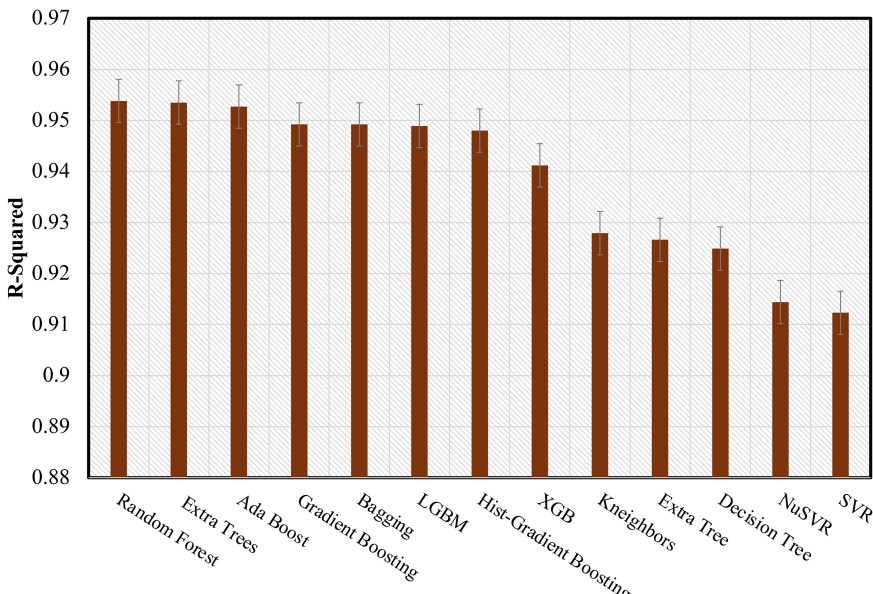

**Fig 5**. **Bar chart of the mean $R^2$ for the best performing regressors.**

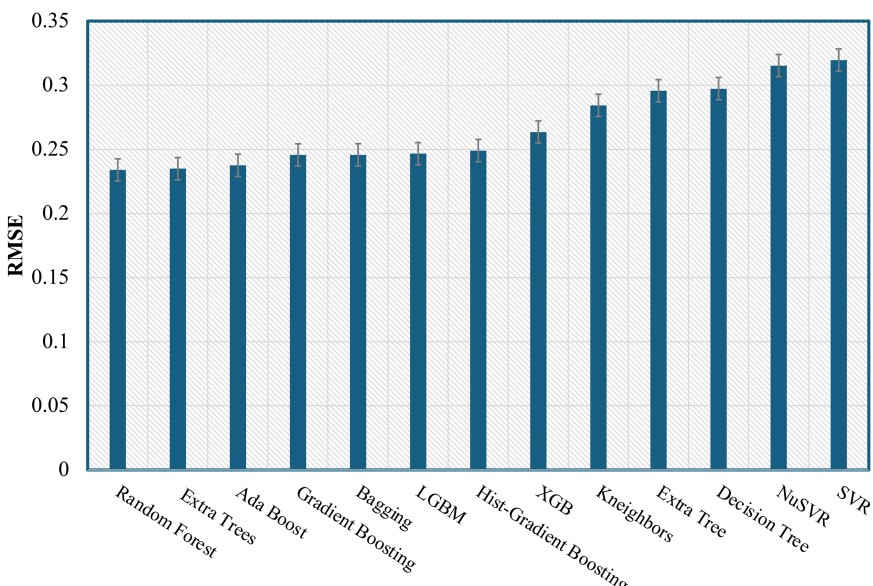

**Fig 6**. **Bar chart of the mean RMSE for the best performing regressors.**

0.01 ms per OFDM block of 100 symbols. This latency is negligible compared to the symbol duration in Li-Fi systems, particularly since the update occurs at the block level rather than per individual symbol. Moreover, inference is inherently parallelizable, enabling efficient deployment on embedded processors or FPGA-based systems. These results indicate that the proposed model achieves a favourable balance between accuracy and efficiency, supporting its feasibility for real-time integration in Li-Fi networks.

### 5.3 Model improvement based on feature selection and hyperparameter tuning

Given the substantial performance of the RF regression model, it is well-suited for future deployment in practical Li-Fi systems. These performance metrics were obtained using the same set of features described in Table 2 of [10] to ensure a fair comparison. However, to further enhance the practicality and interpretability of our model, a feature selection technique was applied to identify the most relevant features for predicting the DC bias. As a result, the Mean and BER features are eliminated due to their weak direct and indirect correlations with the DC bias. The mean value exhibits a weak correlation primarily because the OFDM signal inherently follows a Gaussian distribution with a zero mean. In such a distribution, the overall average of OFDM signal amplitudes is symmetrically distributed around zero regardless of the DC bias level. Consequently, variations in the DC bias do not significantly alter the mean value of the OFDM signal, leading to a weak or negligible correlation between them. Meanwhile, the BER is excluded because the DC bias primarily affects the transmitted signal on the transmitter side of the system, while BER is determined on the receiver side after performing the demodulation process. As a result, including BER in the prediction process would not be practical or meaningful in this context. This procedure of feature refinement is particularly valuable for understanding the impact of each feature on DC bias, which is critical in practical applications. Following this feature selection process, the RF model achieved $R^2 = 0.9436$ and RMSE= 0.2635. The results show a slight decrease in performance compared to the initial features. However, training the model with the most relevant and correlated features improves its generalization and practicability in deployment.

After evaluating the RF model using the most relevant features, a hyperparameter tuning process was employed using the grid search method to find the optimal configuration that maximizes the model's predictive accuracy while ensuring strong generalizability to unseen data. The hyperparameters, tuned values, range of search parameters, and the performance scores are shown in Table 5.

This chosen range of parameters was selected around the bounds of the defaults and typical values known to perform well, while ensuring relevance and computational efficiency. The remaining hyperparameters not included in this table were retained at their default values to maintain practical training without excessive computational cost, as the RF model is known to achieve reliable accuracy with default settings in many applications [45,46]. To ensure that the experimental setup can be fully replicated, a complete table in Appendix A lists all RF hyperparameters, including both the reported tuned values and those retained at their default.

After tuning the hyperparameters with the selected input features, the model performance improved, with $R^2$ increasing to 0.9450 and RMSE decreasing to 0.2603, as shown in Table 6. Although this represents only a modest improvement, it constitutes an important and necessary step. The DC bias is strongly correlated with the BER feature, which accounts for the slightly higher performance when BER is included. As mentioned earlier in this section, in practical transmitter-side DC

**Table 5. Results of hyperparameter tuning for RF regressor.**

| Hyperparameter | Search Parameters | Tuned Value |
|---|---|---|
| bootstrap | [True, False] | True |
| max depth | [None, 10, 20, 30] | None |
| max features | ['auto', 'sqrt', 'log2'] | auto |
| min samples leaf | [1, 2, 4] | 4 |
| min samples split | [2, 5, 10] | 2 |
| n estimators | [50, 100, 200] | 200 |

**Table 6. Performance of RF regression model.**

| RF Model Evaluation | $R^2$ | RMSE |
|---|---|---|
| Model performance with "Mean" and "BER" features | 0.9538 | 0.2339 |
| Model performance after feature selection | 0.9436 | 0.2635 |
| Model performance after feature selection and tuned parameters | 0.9450 | 0.2603 |

bias optimization, incorporating BER as a predictor is not feasible. Excluding it, therefore, ensures that the model delivers strong and reliable performance suitable for real-world deployment.

## 5.4 Superiority of RF model over polynomial regression

In reference [10], data samples were analyzed using linear and polynomial regression models to map the features of the OFDM signal to the optimized DC bias. The reported performance metrics were $R^2 = 0.8406$, RMSE = 0.4279 for linear regression, and $R^2 = 0.9677$, RMSE = 0.1925 for polynomial regression, suggesting that the polynomial model is the best-performing approach for DC bias prediction in DCO-OFDM systems. To further evaluate the robustness of the polynomial model, we re-assessed its performance on same data but shuffled samples. This resulted in a notable decline in performance, with $R^2$ dropping to 0.8922 and RMSE increasing to 0.3595. In contrast, the RF model developed in this study achieved a consistent $R^2 = 0.9450$ and RMSE = 0.2603, demonstrating superior accuracy and performance compared to the polynomial. Table 7 summarizes the performance improvement in the RF model relative to the results reported in [10].

The observed performance reduction in the polynomial model appears to rely on the sequential pattern present in the original dataset. This scenario is unlikely to occur in practical transmission where the system parameters dynamically vary in response to changing channel conditions rather than following a predetermined order. The RF model, however, maintained robust performance on both the original and shuffled datasets. As shown in Fig 7, the proposed RF-based

**Table 7. Performance comparison between Ref [10] and the proposed RF model.**

| Model | $R^2$ | RMSE | Remarks |
|---|---|---|---|
| Polynomial Regression in [10] | 0.9677 | 0.1925 | Given as the best performance |
| Polynomial Regression after data shuffling | 0.8922 | 0.3595 | Performance degradation: ↓7.81% in $R^2$, ↓86.75% in RMSE |
| RF in this work for the same features in [10] | 0.9538 | 0.2339 | Improved performance compared to the results with data shuffling: ↑6.91% in $R^2$, ↑34.93% in RMSE |
| Final RF model compared to polynomial | 0.9450 | 0.2603 | Improved performance and robustness after features refinement and tuning: ↑5.92% in $R^2$, ↑27.59% in RMSE |

*Note:* ↑ indicates improvement (increase in $R^2$ or decrease in RMSE), while ↓ indicates reduction in performance.

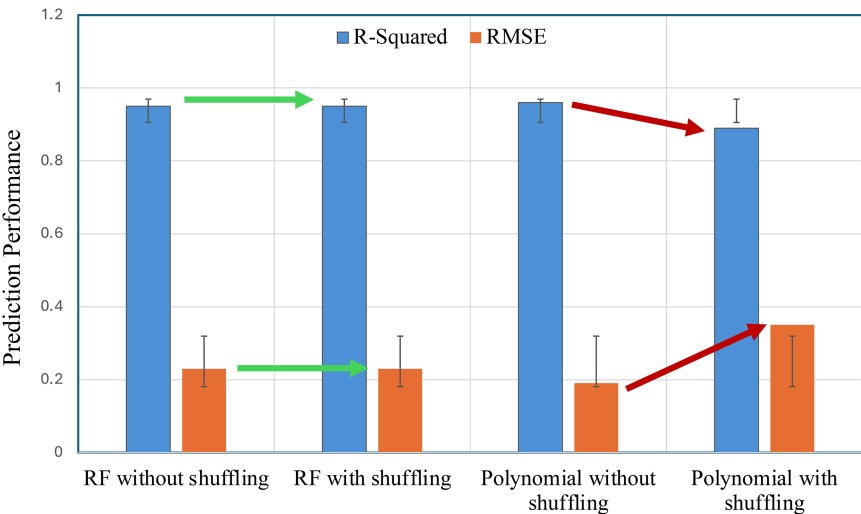

**Fig 7**. **Performance comparison of RF model with polynomial regression (with and without data shuffling).**

approach provides a meaningful improvement and incremental contribution over prior polynomial model. Thus, highlighting its reliability and suitability for deployment in real-world DCO-OFDM systems.

## 5.5 Model improvement based on increasing data samples

The dataset used in this study consists of 250 OFDM samples, which is a legitimate number for experimental work in wireless communication. Increasing the number of training samples was investigated using data augmentation methods to enhance the performance and robustness of the ML model. In methods such as generative adversarial networks, highly realistic data samples that closely resemble the training data can be generated. However, this method is particularly effective for augmenting image data [47]. Similarly, the synthetic minority over-sampling technique can generate new synthetic data by interpolating between the existing training samples.

In the context of this study, the OFDM signal is different in nature from typical data used in these augmentation techniques. It has unique temporal and spectral properties where the orthogonality of sub-carriers is critical to prevent interference. Additionally, the phase and amplitude of the signal are sensitive to noise due to the FFT modulation/ demodulation process. Therefore, applying these standard data augmentation methods to OFDM signals is inadvisable because it can distort the real representation of the properties of the transmitted signal. In this research dataset, the transmission characteristics, denoted by N and M, shape the statistical properties of the transmitted signal and thereby the required DC bias for the corresponding signal. The number of samples for each M and N was not evenly distributed during the simulation, giving insufficient representation to each corresponding signal transmission case.

Bootstrap sampling is a powerful technique used to address the challenges posed by small datasets. In this technique, new data samples are generated by repeatedly drawing samples from the original dataset with replacements. By applying this technique to increase the dataset size, a significant improvement in the RF model performance was achieved, as shown in Table 8. The performance of $R^2$ and RMSE improved when the data was doubled, achieving 0.9776 and 0.1626, respectively. Further improvement was observed when the dataset size was tripled, with $R^2$ reaching 0.9938 and RMSE decreasing to 0.0851. This significant improvement resulted from the redundancy of the new samples involved in the testing set. This technique provides a significant improvement in balancing the uneven distribution of data and can be applied more effectively in classification problems than in regression to create class balance.

## 5.6 Limitations

This study acknowledges certain limitations that also present future opportunities for research. Although the dataset used introduces a variety of transmission characteristics and captures key statistical features relevant to DC bias optimization, one limitation is its relatively small size of 250 OFDM symbols. The dataset is also not evenly distributed, with certain (N, M) pairs being more heavily represented than others. While sufficient for a proof-of-concept demonstration, a larger dataset would further enhance the robustness and generalizability of the results. Future work could focus on incorporating a larger number of training samples while ensuring an even distribution of key parameters. Additionally, the training process could be supported by employing statistical measures such as the Kappa index to validate the adequacy of the chosen dataset size in relation to the model's performance [48]. Addressing these aspects would further enhance the scalability, reliability, and broader applicability of the proposed methodology.

**Table 8. Performance of RF regressor with different data size.**

| Data Size | $R^2$ | RMSE |
|---|---|---|
| 250 | 0.9450 | 0.2603 |
| 500 | 0.9776 | 0.1626 |
| 750 | 0.9938 | 0.0851 |

# 6 Conclusion

In this paper, a robust ML selection process was developed to predict the optimized DC bias in DCO-OFDM transmission schemes. The optimal DC bias was determined using the transmitted OFDM signal features to mitigate the impact of clipping noise, thereby improving the overall transmission performance in Li-Fi systems. This study employed a robust ML regression algorithm, which incorporates LPA, to evaluate the performance of various ML regression models. The process of model training was iterated to statistically validate the results. The simulation results demonstrated that the RF regression model outperformed the findings of the previous study. The RF model exhibited strong performance through a comprehensive training and evaluation process, showing its ability to generalize and accurately predict the DC bias in practical applications. Further model improvement, such as increasing the dataset size and hyperparameter tuning, was conducted to ensure the model's robustness and stability. The developed model is a highly suitable candidate for real-world deployment in DCO-OFDM systems, offering a substantial improvement over conventional optimization methods.

## Supporting information

**S1 Appendix. Table 9. Complete list of RF hyperparameters.**
(PDF)

## Acknowledgments

This research forms part of the doctoral work of the first author, supervised by the second and third authors at the College of Science and Engineering (CSE) at the University of Leicester (UOL), Leicester, United Kingdom.

## Author contributions

**Conceptualization:** Marwah Salman.

**Formal analysis:** Marwah Salman, David Siddle, Yuan Gao.

**Methodology:** Marwah Salman, Yuan Gao.

**Software:** Marwah Salman.

**Supervision:** David Siddle, Yuan Gao.

**Visualization:** Marwah Salman, Yuan Gao.

**Writing – original draft:** Marwah Salman.

**Writing – review & editing:** David Siddle, Yuan Gao.

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
