## [Decision Letter · Decision Letter 0]

30 Dec 2024

PONE-D-24-52879A robust ML regressor model for predicting the DC bias in

DCO-OFDM based Li-Fi technologyPLOS ONE

Dear Dr. Salman,

Thank you for submitting your manuscript to PLOS ONE. After careful consideration, we feel that it has merit but does not fully meet PLOS ONE’s publication criteria as it currently stands. Therefore, we invite you to submit a revised version of the manuscript that addresses the points raised during the review process.

We look forward to receiving your revised manuscript.

Kind regards,

Khalid Taher Mohammed Al-Hussaini, Ph.D

Academic Editor

PLOS ONE

Journal Requirements:

4. We note that your Data Availability Statement is currently as follows: “All relevant data are within the manuscript and in Supporting Information files.”

Reviewers' comments:

Reviewer's Responses to Questions

**Comments to the Author**

1. Is the manuscript technically sound, and do the data support the conclusions?

Reviewer #1: Partly

Reviewer #2: Partly

2. Has the statistical analysis been performed appropriately and rigorously? 

Reviewer #1: No

Reviewer #2: Yes

3. Have the authors made all data underlying the findings in their manuscript fully available?

Reviewer #1: No

Reviewer #2: No

4. Is the manuscript presented in an intelligible fashion and written in standard English?

Reviewer #1: Yes

Reviewer #2: Yes

5. Review Comments to the Author

Reviewer #1: 1-Please rewrite the paper abstract to be more directive.

2-Please provide the full definition of any abbreviation upon its first appearance such as ML in line 7.

3-The dataset used (250 samples) is relatively small, which limits the robustness and generalizability of the results. Please use a larger data set.

4-Uneven distribution of transmission characteristics (e.g., N and M) in the dataset could lead to biases in the trained model.

5-While some features (e.g., BER) were excluded for practical reasons, the paper does not explore the potential inclusion of new, alternative features that could enhance prediction accuracy?

6-There is limited discussion about the interpretability of the chosen features, particularly in terms of their physical relevance to Li-Fi systems.

7-The study focuses on offline data and does not address how the model can be integrated into a real-time Li-Fi system where computational resources and latency are critical factors.

8-The assumption that Random Forest is well-suited for practical deployment is made without benchmarking its computational efficiency.

9-The model evaluation does not account for varying channel conditions (e.g., multipath fading, environmental interference). The noise model is limited to additive white Gaussian noise (AWGN).

10-Although grid search was used for hyperparameter tuning, the paper lacks a discussion of the computational trade-offs and the reasoning behind the selected parameter ranges.

11-The performance drop observed after data shuffling suggests that prior models may have exploited patterns specific to the dataset. Please explore why such patterns existed and whether they might arise in other datasets.

12-While the study emphasizes the importance of optimizing DC bias for DCO-OFDM, it does not place this work in the broader context of competing Li-Fi technologies or how these results might advance the field overall.

13-The quality of all the figures in the paper is quite low especially, Fig. 4.

14-Please ensure that all sections of the paper are numbered and integrate the figures directly within the text at appropriate locations.

15-English proofreading is necessary.

Reviewer #2: The manuscript examines the use of machine learning techniques to predict the DC bias in DCO-OFDM systems employed in Li-Fi technology.

The study presents a machine learning method for predicting the DC bias in DCO-OFDM systems associated with Li-Fi technology. However, several scientific shortcomings and methodological issues undermine the robustness of the findings and the novelty of the investigation. Comments and suggestions for the authors include:

1. Equation (1): The definition of the inverse fast Fourier transform (IFFT) operation must be clearly articulated. An explanation of how the Gaussian distribution correlates with the signal properties and influences transmission performance should be included.

2. Equation (2): A more detailed explanation of the definition of BDC is necessary. Discuss how altering the scaling factor (µ) affects the performance of the transmitted signal, particularly regarding clipping noise.

3. Investigate how variations in key parameters such as signal-to-noise ratio (SNR) or channel conditions influence the model’s predictions. Have sensitivity analyses been conducted to quantify these impacts?

4. System Model Description: The description of the system model is unclear and lacks sufficient detail, which prevents the reader from understanding the implemented mechanisms.

5. The manuscript does not provide empirical verification of experimental results against simulated outcomes. Validation of experimental data with simulation results is crucial for establishing the model's efficacy and reliability.

6. The research introduces several machine learning techniques; however, it does not sufficiently distinguish itself from existing studies, especially Reference [8], where similar methodologies were applied. A stronger focus on how this study advances the field compared to prior work is required to highlight its originality and significance.

7. To reduce the risk of overfitting and enhance the robustness of the machine learning model, implementing cross-validation techniques during training is recommended.

8. While including the Friedman test is commendable, it is suggested to complement this with other statistical methods (such as paired t-tests or ANOVA) to further validate the findings and assess the significance of differences between model performances.

9. A more in-depth discussion regarding the transition of the proposed model into practical applications within Li-Fi systems would increase the manuscript’s relevance and impact.

10. The manuscript notes a decline in the performance of the polynomial regression model when the data samples are shuffled. This raises considerable concerns about the model's generalizability. The authors should clarify the measures taken to ensure robustness against variations in data distribution and discuss the implications for real-world applications.

11. Although the paper mentions a comprehensive feature analysis, it lacks a clear justification for the relevance and importance of the selected features. This absence raises questions about potential omissions that might affect the model's predictive accuracy.

12. References: The references require updates to include recent 2024 studies for stronger support, as current references do not sufficiently reinforce relevance.

- https://doi.org/10.1016/j.optcom.2024.130558

- https://doi.org/10.1007/s11082-020-02497-0

- https://doi.org/10.1109/ACCESS.2019.2924531

- https://doi.org/10.1016/j.optcom.2018.12.034

---

6. PLOS authors have the option to publish the peer review history of their article (what does this mean?). If published, this will include your full peer review and any attached files.

Reviewer #1: No

Reviewer #2: **Yes: **Ebrahim E. Elsayed

---

## [Author Response · Author response to Decision Letter 1]

26 Mar 2025

Response to Reviewer 1

Reviewer #1, Concern# 1 (Please rewrite the paper abstract to be more directive.)

Author response: Thank you for your feedback. We have carefully revised the abstract to make it more directive, outlining the research objectives, methods, and key findings. The manuscript’s abstract has been updated accordingly (Line 2-20, Page 1).

Reviewer #1, Concern# 2 (Please provide the full definition of any abbreviation upon its first appearance such as ML in line 7.)

Author response: Thank you for raising this concern. We have ensured that all abbreviations are defined upon their first appearance in the manuscript (Line 4, Page 1, and Line 45, Page 2).

Reviewer #1, Concern# 3 (The dataset used (250 samples) is relatively small, which limits the robustness and generalizability of the results. Please use a larger data set.)

Author response: Thank you for bringing up the importance of dataset size. We response to this as follows:

• The dataset used in this study contains 250 individual OFDM symbols, which was previously published in the PLOS One journal, 2021. The dataset is associated with the article titled “Machine learning for DCO-OFDM based Li-Fi” and is accessible via the DOI: https://doi.org/10.1371/journal.pone.0259955.s001. While this dataset size may appear limited to most of the ML works, it is sufficient to capture a wide range of signal features variations relevant for addressing the DC optimization problem. For each OFDM symbol, that paper extracted crucial statistical features of the transmitted signal which significantly influence the DC bias required to keep the transmission reliability within a specific BER.

• We believe that the validation and testing of the model were already conducted using the training and testing splits. These splits, combined with repeated runs (five iterations in our case) and statistical validation using the Friedman test to ensure the robustness and generalizability of our model. Using this test, we aim to determine the significant differences among the results obtained from each iteration on the data. Meanwhile, to reduce the risk of potential bias in the developed model, a random shuffle of the data is undertaken with each iteration. This prevents certain patterns in the original ordered data from being learned, thus training the model to learn more generalized patterns rather than particular pattern related to the order of data samples. Therefore, the mean values of the performance scores were considered to select the best ML regressor that fits the given dataset (see subsection “Machine learning model selection framework” in Page 8).

• In future work, we will conduct further experiments to assess the impact of the dataset size on machine learning performance. Although we initially considered choosing a relatively small dataset of OFDM symbols, when increasing the number of symbols in a future study, the proposed methodology should be totally the same. Therefore, the novelty of this paper has been clearly delivered. For example, a statistical measure of reliability such as Kappa index can provide an insight into the dataset’s impact on model training. If the Kappa index shows minimal improvement with more data, it suggests that the current dataset size is adequate for training.

We updated the manuscript by mentioning these points in the subsection Data Description (Line 234-264, Page 7), Machine learning model selection framework (Line 265-302, Page 8), Machine learning model evaluation (Line 313- 329, Page 9), and Model training and regression model selection (Line 349- 357, Page 10)

Reviewer #1, Concern# 4 (Uneven distribution of transmission characteristics (e.g., N and M) in the dataset could lead to biases in the trained model.)

Author response: Thank you for raising this concern. It should be noted that the transmission characteristics N and M must be powers of two. The N parameter defines the number of subcarriers used to perform OFDM transmission. However, in this type of transmission, a large number of subcarriers is typically used to ensure that the signal is a Gaussian random variable with zero mean. If the amplitude of these signals follows the Gaussian distribution means that 97% of these amplitudes lie within the range of twice the standard deviation of the signal. In this dataset the M values of 256, 512, 1024 were considered.

While the M parameter defines the modulation order or the mapping level of the QAM modulation. The larger M the more the signal is susceptible to channel noise. In this study, the N values considered are 4, 16, 64, 256, 1024. The values taken by N and M in Ref [9] aggregated all the values considered in the literature. We believe that an even distribution of these values would support the model robustness (as discussed in subsection “Data description” Line 248-255, Page 7).

Reviewer #1, Concern# 5 (While some features (e.g., BER) were excluded for practical reasons, the paper does not explore the potential inclusion of new, alternative features that could enhance prediction accuracy?)

Author response: Thank you for bringing up this important point. In the manuscript, we mentioned that the BER feature was excluded because the DC optimization problem primarily impacts transmission performance by introducing a clipping noise at the transmitter. This type of noise is independent of other channel noise as it occurs before the signal passes through the channel. Therefore, we believe that addressing this issue at the transmission side is the most effective approach to handle the clipping noise.

While we acknowledge the potential value of exploring alternative features, our primary objective in this work was to improve prediction accuracy for the specific dataset originally introduced by Ref [9] in 2022. Their piece of work presented a new ML-based approach for DC bias optimization in DCO-OFDM systems, which had not been previously explored. Building on this, we focused on enhancing the performance of the ML model using the same dataset without introducing new features. Instead, we concentrated on refining the feature set by excluding features that don’t positively correlate to the DC bias or could negatively affect the model's performance.

Furthermore, the updated feature set was then used to evaluate the model's performance, employing the proposed regression algorithm to identify the best-performing ML regressor for the dataset. These findings are discussed separately in the subsection Model improvement based on feature selection and hyper-parameter tuning (Line 382-417, Page 13), where we demonstrate how ML techniques, such as feature selection and hyper-parameter tuning, significantly influenced the model's performance.

Reviewer #1, Concern# 6 (There is limited discussion about the interpretability of the chosen features, particularly in terms of their physical relevance to Li-Fi systems.)

Author response: Thank you for highlighting this concern. In the updated manuscript, we have added the discussion on the interpretability of the selected features and their physical relevance to Li-Fi systems. Specifically, we have provided a detailed explanation of how each feature influences DC bias optimization in DCO-OFDM transmission, and their role in transmission performance. Additionally, as our study builds upon the work presented in Ref [9], we note that a comprehensive discussion of the chosen features has already been provided there, and we have explicitly referenced that in our revised manuscript.

We have updated the manuscript by including a detailed description of the dataset in subsection Data description in (Line 235-264).

Reviewer #1, Concern# 7 (The study focuses on offline data and does not address how the model can be integrated into a real-time Li-Fi system where computational resources and latency are critical factors.)

Author response: Thank you for raising this concern. In our current work, the primary objective was to develop an accurate ML model that can predict the optimum DC bias as a proof of concept. The robust selection process employed in our study was mainly designed to identify the most effective ML model for the DC bias prediction task. Once these simulation results are validated, the proposed model can be integrated in the real-time Li-Fi applications. The proposed model can continuously receive real-time data and predict the optimum DC bias parameter for the optical transmitter, the method should be the same. However, that would be a future study since it is not realistic to finish during this revision. Thank you for your understanding.

To ensure computational efficiency and minimize latency, a block-based approach can be adopted rather than making per-symbol predictions. The model can aggregate statistical features from multiple OFDM symbols to determine a suitable DC bias that accommodates their PAPR variations. This approach significantly reduces the update frequency, making real-time deployment more feasible given the short duration of individual OFDM symbols. It is worth mentioning that the application of ML to optical wireless networks is still progressing. However, this paradigm is increasingly recognized as a promising solution to growing developments and to the need to optimize the networks’ parameters.

We have updated the manuscript by mentioning our response to this point in the following subsections:

• Machine learning in practical Li-Fi applications (Line 213- 230, Page 6)

• Machine learning model selection framework (Line 265-302, Page 8)

Reviewer #1, Concern# 8 (The assumption that Random Forest is well-suited for practical deployment is made without benchmarking its computational efficiency.)

Author response: We acknowledge the concerns regarding the computational efficiency of the RF model. Our approach is that the ML model is trained offline, that means the computational burden occurs during training stage, with no impact on real-time system performance. Once the model deployed, the system only needs to evaluate the signal's statistical properties and transmission characteristics to perform prediction and determine the optimum DC bias using a trained model, eliminating further complex computations.

In addition, the RF model is inherently parallelizable due to the training of individual internal weak learners (decision trees), allowing for efficient execution on low-power embedded processing units if the model can be optimized during the training.

We updated the manuscript by modifying the “Model training and regression model selection” subsection (Line 360-379, Page 12), making a clear statement regarding this concern.

Reviewer #1, Concern# 9 (The model evaluation does not account for varying channel conditions (e.g., multipath fading, environmental interference). The noise model is limited to additive white Gaussian noise (AWGN).)

Author response: Thank you for bringing up this concern. In Ref [9], the BER target for DCO-OFDM simulation was established based on the DC bias optimization process at the transmitter-side. This optimization primarily aimed to mitigate the impact of clipping noise, ensuring that the BER remained within the acceptable range before transmission. However, once the signal propagates through the channel, additional noise sources, such as AWGN, multipath fading, and background interference, can further degrade the signal quality. The reason is that their simulation model was limited to AWGN because it is the dominant noise source in indoor optical communication environments.

Note that the chosen BER benchmark was specifically designed to confine the clipping impairment at the transmitter to an acceptable level and addressing it separately from any other type of noise in the channel, where the latter can be effectively addressed through channel estimation and equalization techniques at the receiver. Our study focuses on the ML- driven aspects of the DC optimization process, particularly features analysis, model selection, and performance evaluation, using the dataset in Ref [9].

We updated the abstract of the manuscript to clarify the focus of our study as well as the key findings by mentioning our response to this concern in subsection Data description (Line 235-242, Line 256-260, Page 7).

Reviewer #1, Concern# 10 (Although grid search was used for hyperparameter tuning, the paper lacks a discussion of the computational trade-offs and the reasoning behind the selected parameter ranges.)

Author response: Thank you for highlighting this concern. This selected range of hyperparameters was chosen around the bounds of well-established defaults and commonly known values that balance model performance and computational efficiency. The aim was to explore a certain range of parameters that is both effective and practical without excessive computations. While the rest of hyperparameters not tuned were kept at their default values to maintain the model stability. Given that our model training was performed offline, this approach was feasible without impacting real-time system performance.

We updated the manuscript by clarifying this concern in subsection Model improvement based on feature selection and hyperparameter tuning in (Line 401 – 417, Page 14).

Reviewer #1, Concern# 11 (The performance drop observed after data shuffling suggests that prior models may have exploited patterns specific to the dataset. Please explore why such patterns existed and whether they might arise in other datasets.)

Author response: Thank you for commenting on the observed performance drop. Our initial analysis indicates that the dataset produced by Purnita et al in Ref [9] inherently included an ordering due to the experiment setup during the simulation. The data collection process systematically varied the constellation size of QAM modulation for each subcarrier. These systematic changes of the key transmission parameters (M and N) introduced a predictable pattern, which was exploit by the polynomial ML model during the training phase, therefore high-performance metrics were achieved in their evaluation. In real-world scenarios, such structured changing to the parameters is unlikely occur, as the transmission parameters typically change in response to dynamic channel conditions rather than following a predetermined order. Therefore, in our study, we maintained the practice of shuffling the data samples for each iteration run to ensure that our model's performance is robust and truly applicable in real-word systems.

We updated the manuscript by changing the title of the subsection “Impact of data shuffling on model performance “to “RF Model’s accuracy and robustness”, and we added a brief explanation to clarify this point (Line 451-473, Page 16).

Reviewer #1, Concern# 12 (While the study emphasizes the importance of optimizing DC bias for DCO-OFDM, it does not place this work in the broader context of competing Li-Fi technologies or how these results might advance the field overall.)

Author response: Thank you for addressing this concern. The superiority of the proposed model lies in its ability and generality to optimize the DC bias in DCO-OFDM system based on the transmission characteristics and the signal properties. Unlike the conventional methods that employ a fixed DC bias, which is inefficient practice for DCO-OFDM transmission. The proposed model adjusts the DC bias with the help of ML model trained according to the required data rates or the channel conditions. The trained model in our approach covers a wide range of transmission settings that have not been integrated in this manner before and could be also used to solve the optimization problem in non-adaptive DCO-OFDM system due to the generality of our model. A sufficient DC bias is required to ensure reliable and efficient transmission for low data rates. But as the demand for higher data rates increases, the model is capable of predicting the optimum DC bias across varying conditions to maintain the BER within the required benchmark.

The trained ML model outperformed the ML model in literature that utilized polynomial regression to predict the DC bias. There is a wide range of ML models to choose from, but yet, no theore

---

## [Decision Letter · Decision Letter 1]

5 Aug 2025

PONE-D-24-52879R1A robust ML regressor model for predicting the DC bias in

DCO-OFDM based Li-Fi technologyPLOS ONE

Dear Dr. Salman,

Thank you for submitting your manuscript to PLOS ONE. After careful consideration, we feel that it has merit but does not fully meet PLOS ONE’s publication criteria as it currently stands. Therefore, we invite you to submit a revised version of the manuscript that addresses the points raised during the review process.

 Several reviews have been received for your revised version of the paper. While the general impression is that the manuscript has improved, some reviewers think that you have not adressed the original comments in a completely satisfactory way. I agree with them so I think another round of revision is needed. Some of the reviewers also give interesting suggestions for improving the presentation of the paper. Please, follow them carefully. Additionally, one of the reviewers has commented to me that "the authors should add at least one quantitative benchmark (e.g., vs. XGBoost or conventional methods) to strengthen claims of superiority. Without this, the work risks appear incremental". I would like you to follow this advice or, at least, to know your opinion about it.  Please submit your revised manuscript by Sep 19 2025 11:59PM. If you will need more time than this to complete your revisions, please reply to this message or contact the journal office at plosone@plos.org. Please include the following items when submitting your revised manuscript:

We look forward to receiving your revised manuscript.

Kind regards,

Miguel A. Fernández, Ph.D.

Academic Editor

PLOS ONE

Journal Requirements:

1. Thank you for declaring that the code underpinning this study will be made available on request. In order to comply with the PLOS One policy on code sharing, please review our guidelines at https://journals.plos.org/plosone/s/materials-and-software-sharing#loc-sharing-code and before submitting your revised manuscript please ensure that your code is shared in a way that follows best practice and facilitates reproducibility and reuse. We thank you for your attention to this request.

Reviewers' comments:

Reviewer's Responses to Questions

**Comments to the Author**

1. If the authors have adequately addressed your comments raised in a previous round of review and you feel that this manuscript is now acceptable for publication, you may indicate that here to bypass the “Comments to the Author” section, enter your conflict of interest statement in the “Confidential to Editor” section, and submit your "Accept" recommendation.

Reviewer #1: (No Response)

Reviewer #2: (No Response)

Reviewer #3: All comments have been addressed

Reviewer #4: (No Response)

Reviewer #5: All comments have been addressed

2. Is the manuscript technically sound, and do the data support the conclusions?

Reviewer #1: Partly

Reviewer #2: (No Response)

Reviewer #3: Yes

Reviewer #4: Partly

Reviewer #5: Yes

3. Has the statistical analysis been performed appropriately and rigorously? 

Reviewer #1: Yes

Reviewer #2: (No Response)

Reviewer #3: Yes

Reviewer #4: No

Reviewer #5: Yes

4. Have the authors made all data underlying the findings in their manuscript fully available?

Reviewer #1: No

Reviewer #2: (No Response)

Reviewer #3: Yes

Reviewer #4: Yes

Reviewer #5: Yes

5. Is the manuscript presented in an intelligible fashion and written in standard English?

Reviewer #1: Yes

Reviewer #2: (No Response)

Reviewer #3: Yes

Reviewer #4: No

Reviewer #5: Yes

6. Review Comments to the Author

Reviewer #1: The authors have addressed the reviewers' concerns comprehensively, improving the clarity, robustness, and relevance of the manuscript. However, the following concerns still have more work.

1. (Concern #3)

o Author Response: The authors justify the 250-sample dataset by citing prior work and emphasize robustness via shuffling, statistical tests, and future plans for larger datasets.

o Reviewer comment: Adequate for a proof-of-concept study, but the limitation should be explicitly noted in the "Limitations" section of the manuscript.

2. (Concern #6)

o Author Response: Added discussion on physical relevance of features (e.g., BER exclusion due to transmitter-side optimization).

o Reviewer comment: Improved but could briefly summarize key features' roles (e.g., how signal statistics directly influence DC bias) in the "Data Description" section.

3. (Concern #7)

o Author Response: Acknowledged as future work; proposed block-based approach to reduce computational load.

o Reviewer comment: Satisfactory, but a brief computational complexity analysis (e.g., inference time per OFDM block) would strengthen feasibility claims.

4. (Concern #10)

o Author Response: Explained parameter ranges and offline training feasibility.

o Reviewer comment: Clear, but a table or appendix listing all hyperparameters (default vs. tuned) would enhance reproducibility.

5. (Concern #8)

o Author Response: Friedman test used due to non-parametric data; p-values support significance.

o Reviewer comment: Well-justified. Consider adding a sentence on effect sizes to complement p-values.

6. (Concern #12)

o Author Response: Emphasized DCO-OFDM's spectral efficiency but lacked comparison to ACO-OFDM or Flip-OFDM.

o Reviewer comment: The authors should add at least one quantitative benchmark (e.g., vs. XGBoost or conventional methods) to strengthen claims of superiority. Without this, the work risks appear incremental.

Reviewer #2: * The authors' responses to reviewer comments have been weak, and the revisions made are too light. The paper requires more detailed explanations, important updates, and a stronger, more detailed revision to meet the expected level of quality. The authors need to significantly improve the practical discussion, revise the results, and strengthen the link between the results and the discussion. The methodology chosen has clear weaknesses and requires a more thorough explanation. The paper's technical discussion is weak, and the results are redundant and insufficient, with some illogical comparisons in the figures.

*The literature review is lacking and should be significantly expanded to better position the proposed architecture within existing research. The references are outdated, and newer studies from 2024 must be included to support the arguments made in the paper.

https://doi.org/10.1049/ote2.12111

https://doi.org/10.1007/s11082-024-06692-1

https://doi.org/10.1007/s11082-023-05721-9

https://doi.org/10.1007/s12596-024-01929-4

* The abstract is unclear and needs to be edited for better clarity. Additionally, the highlights are overly long and should be shortened to a single line for better impact.

* The English language quality is inadequate, with grammar, sentence structure, and proofreading needing substantial improvement. The overall writing must be clearer and more professional.

* The evaluation model does not account for varying channel conditions, such as multipath fading and environmental interference. The noise model is limited to additive white Gaussian noise (AWGN), which is not sufficient for this type of work.

Reviewer #3: The work improves model prediction performance of a previously published work/dataset by using advanced machine learning tools. Their proposed model proves to be more robust, while enhancing accuracy of DC bias prediction compared to previous approaches.

I have reviewed all the concerns and answers to Reviewer #1 and #2. This have been significantly addressed, and should be accepted with the following concerns addressed.

See Reviewer Comments Attached

Reviewer #4: This manuscript presents a machine-learning (ML)–based regression framework to predict the optimal DC bias in DC-biased optical OFDM (DCO-OFDM) for indoor Li-Fi systems. In my opinion, this manuscript could be improved in the following ways:

1. The mathematical description of the IFFT process and DC bias definition lacks sufficient detail. Please expand on how Hermitian symmetry is enforced and explicitly derive the relationship between the scaling factor µ and the DC bias in both linear and dB domains.

2. The dataset’s uneven distribution of N (subcarriers) and M (QAM order) may bias the model. Provide summary statistics (histograms or tables) showing how many samples correspond to each (N, M) pair, and discuss potential mitigation strategies beyond bootstrap.

3. The work assumes only AWGN noise. Realistic Li-Fi channels often exhibit multipath effects and background light interference. Discuss how these additional impairments could alter the DC bias optimization and whether the model would generalize.

4. The rationale for eliminating Mean and BER is only briefly justified. Please provide quantitative feature-importance scores (e.g., RF feature ranking) and detail any correlations among the retained features to demonstrate the robustness of feature selection.

5. The choice of a single 70/30 split repeated with shuffling is less standard than k-fold cross-validation. Compare the train–test approach with k-fold (e.g., 5-fold) CV to quantify any overfitting risk and to validate the reported performance.

6. While the Friedman test is appropriate, please report the test statistic, degrees of freedom, and exact p-values. Also, consider a post-hoc test (e.g., Nemenyi) to identify which regressors differ significantly.

7. The grid search explored only six RF hyperparameters. Explain why other important parameters (e.g., max_depth, min_impurity_decrease) were kept at defaults, and discuss the computational cost versus performance trade-offs.

8. Furthermore, the work would greatly benefit from including and referencing more recent material on practical implementations of machine learning methods across several domains. It is recommended to engage in discussions on the comparison of results and the integration of these concepts into your work. You may find more information on this topic in the following articles: https://doi.org/10.1007/s11004-023-10116-3, https://doi.org/10.1007/s11069-023-06322-1, https://doi.org/10.1007/s12583-021-1525-9, https://doi.org/10.1007/s12583-021-1407-1.

9. Some figures (e.g., bar charts of R² and RMSE) lack error bars or confidence intervals. Add them to convey statistical variance. Ensure all figures are integrated into the text at appropriate points and are high resolution.

10. The manuscript still contains grammatical errors and inconsistencies (e.g., plural/singular agreement, verb tense). A thorough language edit is needed. Additionally, ensure all sections are numbered and citations follow the journal’s prescribed style.

Reviewer #5: The authors have addressed all the reviewer comments. The authors have reported significant improvement compared to the original submission. The paper may be accepted as is.

7. PLOS authors have the option to publish the peer review history of their article (what does this mean?). If published, this will include your full peer review and any attached files.

Reviewer #1: **Yes: **Emad S. Hassan

Reviewer #2: **Yes: **Ebrahim E. Elsayed

Reviewer #3: No

Reviewer #4: No

Reviewer #5: No

---

## [Author Response · Author response to Decision Letter 2]

12 Oct 2025

Response to Reviewer #1

Comment 1:

The authors justify the 250-sample dataset by citing prior work and emphasize robustness via shuffling, statistical tests, and future plan for larger dataset. Adequate for a proof-of-concept study, but the limitation should be explicitly noted in the "Limitations" section of the manuscript.

Response:

Thank you for your feedback. We explicitly acknowledged this comment in a new subsection to clearly state that for readers. Therefore, our contribution to Purnita’s work is properly framed.

Limitations section has been added to the revised manuscript, the highlighted PDF on page 17, line 510:

Limitations:

This study acknowledges certain limitations that also present future opportunities for research. Although the dataset used introduces a variety of transmission characteristics and captures key statistical features relevant to DC bias optimization, one limitation is its relatively small size of 250 OFDM symbols. While sufficient for a proof-of-concept demonstration, a larger dataset would further enhance the robustness and generalizability of the results. Future work could focus on incorporating a larger number of training samples while ensuring an even distribution of key parameters. Additionally, the training process could be supported by employing statistical measures such as the Kappa index to validate the adequacy of the chosen dataset size in relation to the model’s performance [48]. Addressing these aspects would further enhance the scalability, reliability, and broader applicability of the proposed methodology.

Comment 2:

Added discussion on physical relevance of features (e.g., BER exclusion due to transmitter-side optimization). Improved but could briefly summarize key features' roles (e.g., how signal statistics directly influence DC bias) in the "Data Description" section.

Response:

Thank you for your feedback. We briefly summarized the influence of the statistical features on the DC bias in the data description section.

The following paragraph has been added to the revised manuscript, the highlighted PDF on page 8, line 259:

The statistical features extracted from each waveform provide a compact representation of the signal and directly inform the DC bias adjustment. Specifically, the minimum and maximum values define the signal’s amplitude range, which determines the margin required to avoid clipping distortion. The mean indicates the average offset of the signal, and the standard deviation characterizes the signal’s power distribution and variability, which influences the scaling factor that primarily controls the required bias. The dataset also records the corresponding BER, serving as a performance benchmark. Together, these features enable the prediction of the optimum DC bias necessary to maintain transmission reliability.

Comment 3:

Acknowledged as future work; proposed block-based approach to reduce computational load. Satisfactory, but a brief computational complexity analysis (e.g., inference time per OFDM block) would strengthen feasibility claims.

Response:

Thank you for your feedback. We have updated the manuscript by adding computational complexity section to briefly explaining that to the readers.

The following section has been added to the revised manuscript, the highlighted PDF on page 13, line 390:

Computational Complexity

In our study, all simulations and model training were performed on Apple MacBook M2 Pro chip (10-core CPU, 16-core GPU, 32 GB RAM). It is important to emphasize that the primary computational burden lies in the offline training phase; once the model is deployed, only inference is required in real-time operation. The selected RF regressor is computationally lightweight at the inference stage, as it only requires the system to provide the key parameters for predicting the optimum DC bias. When the model was tested for a block-based approach, the average inference time was approximately 0.01 ms per OFDM block of 100 symbols. This latency is negligible compared to the symbol duration in Li-Fi systems, particularly since the update occurs at the block level rather than per individual symbol. Moreover, inference is inherently parallelizable, enabling efficient deployment on embedded processors or FPGA-based systems. These results indicate that the proposed model achieves a favourable balance between accuracy and efficiency, supporting its feasibility for real-time integration in Li-Fi networks.

Comment 4:

Explained parameter ranges and offline training feasibility. Clear, but a table or appendix listing all hyperparameters (default vs. tuned) would enhance reproducibility.

Response:

Thank you for your suggestion. To improve reproducibility, we have added a complete table in the Appendix that lists all Random Forest hyperparameters, including both the tuned values (as reported in Table 4) and those retained at their default settings.

The following table has been added in the Appendix section of the revised manuscript, the highlighted PDF on page 18, line 550:

Appendix A. Complete Random Forest Hyperparameters

Hyperparameter Default value (scikit-learn) Tuned value in this study

bootstrap True True

criterion "squared_error" squared_error (default)

max_depth None None

max_features 1.0 ("auto") Auto

max_leaf_nodes None None (default)

max_samples None None (default)

min_impurity_decrease 0.0 0.0 (default)

min_samples_leaf 1 4

min_samples_split 2 2

min_weight_fraction_leaf 0.0 0.0 (default)

n_estimators 100 200

n_jobs None None (default)

oob_score False False (default)

random_state None None (default)

verbose 0 0 (default)

warm_start False False (default)

ccp_alpha 0.0 0.0 (default)

max_samples None None (default)

Comment 5:

Friedman test used due to non-parametric data; p-values support significance. Well-justified. Consider adding a sentence on effect sizes to complement p-values.

Response:

Thank you for your feedback. We have updated this section to consider complementing p-values with Friedman statistics and effect sizes.

The following paragraph has been added to the revised manuscript, the highlighted PDF on page 11, line 360:

This evaluation compares the performance of the regressors across five iterations of the model training process. Figure 6 presents the Friedman statistics (right axis) alongside the corresponding p-values (left axis) for each iteration. From the second iteration onward, the Friedman test yields p-values below 0.05, indicating statistically significant performance differences among the regressors. To complement the p-values, the effect sizes, expressed as epsilon-squared, ranged from 0.90 to 1.00, reflecting a very strong proportion of variance explained by these differences. Taken together, these results demonstrate that the regressors differ in a statistically significant manner, and that the observed differences are also substantial and practically meaningful.

Comment 6:

Emphasized DCO-OFDM's spectral efficiency but lacked comparison to ACO-OFDM or Flip-OFDM. The authors should add at least one quantitative benchmark (e.g., vs. XGBoost or conventional methods) to strengthen claims of superiority. Without this, the work risks appear incremental.

Response:

We appreciate the reviewer’s suggestion. We would like to clarify that the primary focus of this work is not to compare different OFDM variants (e.g., DCO-OFDM vs. ACO-OFDM or Flip-OFDM) in terms of spectral efficiency, but rather to develop and evaluate a machine learning framework for predicting the DC bias within DCO-OFDM systems. DCO-OFDM outperforms these variants in terms of spectral efficiency and has well-documented advantages in the literature.

In this study, we compared the proposed Random Forest regressor to the polynomial model reported by Purnita et al. [10], which represents the most relevant prior work in DC bias prediction. As detailed in section “RF Model’s Accuracy and Robustness”, our results show that the RF model achieves higher accuracy and robustness compared to the lower performance achieved by the polynomial algorithm. We believe that our work results appear incremental as this comparison demonstrates the superiority of our ML framework in the intended context.

The “RF model’s accuracy and robustness” section has been updated (both title and content) in the revised manuscript. Please see the highlighted PDF on page 15, line 456:

5.4 Superiority of RF Model over Polynomial Regression

In reference [10], data samples were analysed using linear and polynomial regression models to map the features of the OFDM signal to the optimized DC bias. The reported performance metrics were R^2 = 0.8406, RMSE = 0.4279 for linear regression, and R^2 = 0.9677, RMSE = 0.1925 for polynomial regression, suggesting that the polynomial model is the best-performing approach for DC bias prediction in DCO-OFDM systems. To further evaluate the robustness of the polynomial model, we re-assessed its performance on same data but shuffled samples. This resulted in a notable decline in performance, with $R^2$ dropping to 0.8922 and RMSE increasing to 0.3595. In contrast, the RF model developed in this study achieved a consistent R^2 = 0.9450 and RMSE = 0.2603, demonstrating superior accuracy and performance compared to the polynomial. Table [7] summarizes the performance improvement in the RF model relative to the results reported in [10].

The observed performance reduction in the polynomial model appears to rely on the sequential pattern present in the original dataset. This scenario is unlikely to occur in practical transmission where the system parameters dynamically vary in response to changing channel conditions rather than following a predetermined order. The RF model, however, maintained robust performance on both the original and shuffled datasets. As shown in Fig. 7, the proposed RF-based approach provides a meaningful improvement and incremental contribution over prior polynomial model. Thus, highlighting its reliability and suitability for deployment in real-world DCO-OFDM systems.

Response to Reviewer #2

Comment 1:

The authors' responses to reviewer comments have been weak, and the revisions made are too light. The paper requires more detailed explanations, important updates, and a stronger, more detailed revision to meet the expected level of quality. The authors need to significantly improve the practical discussion, revise the results, and strengthen the link between the results and the discussion. The methodology chosen has clear weaknesses and requires a more thorough explanation. The paper's technical discussion is weak, and the results are redundant and insufficient, with some illogical comparisons in the figures.

Response:

We thank the reviewer for this observation. While we believe that our initial responses addressed the comments concisely, we acknowledge that some points raised in this comment are broad (e.g., “more detailed explanations,” “important updates,” “a more thorough explanation”), which makes it challenging to identify the specific areas of concern. In the current revision, we have carefully updated the manuscript in line with the reviewers’ feedback to ensure that the explanations are clearer, the revisions are more substantial, and the contributions of our work are better articulated. We hope that these changes sufficiently address this comment.

Comment 2: The literature review is lacking and should be significantly expanded to better position the proposed architecture within existing research. The references are outdated, and newer studies from 2024 must be included to support the arguments made in the paper.

https://doi.org/10.1049/ote2.12111

https://doi.org/10.1007/s11082-024-06692-1

https://doi.org/10.1007/s11082-023-05721-9

https://doi.org/10.1007/s12596-024-01929-4

Response:

We thank the reviewer for highlighting the importance of including up-to-date literature. We would like to clarify that our manuscript does incorporate recent studies: 33 of our cited works are from 2020–2025, 11 from 2015–2019, and only a small number from 2011–2015. This means that approximately 65% of our references are drawn from the most recent six years, ensuring that our work is well-grounded in current advances.

We also note that many of the suggested studies are focused on free-space optical (FSO) communication systems, such as WDM, MIMO-FSO, and OCDMA architectures, typically applied to outdoor, long-range links. These studies are do not directly address our focus, which is the application of machine learning to optimize DC bias in DCO-OFDM-based Li-Fi transmission systems.

Comment 3:

The abstract is unclear and needs to be edited for better clarity. Additionally, the highlights are overly long and should be shortened to a single line for better impact.

Response:

We thank the reviewer for this comment. We respectfully believe that the current abstract is already clear and coherent, as it concisely introduces the problem, explains the motivation, outlines the methodology, and highlights the key findings of our study. We also note that no other reviewers expressed concerns regarding the clarity of the abstract.

Regarding the highlights, we have intentionally included the changes in the response to reviewer file in this revision and structured them to provide concise informative points. We believe the current format provides the necessary clarity and impact.

Comment 4:

The English language quality is inadequate, with grammar, sentence structure, and proofreading needing substantial improvement. The overall writing must be clearer and more professional.

Response:

We thank the reviewer for this feedback. We would like to note that the manuscript has been carefully proofread, including by a native English speaker (the second author). We have undertaken an additional round of thorough language editing to further improve the manuscript. We believe the revised manuscript is now clearer for both native and non-native English readers.

Comment 5:

The evaluation model does not account for varying channel conditions, such as multipath fading and environmental interference. The noise model is limited to additive white Gaussian noise (AWGN), which is not sufficient for this type of work.

Response:

We appreciate the reviewer’s concern regarding channel impairments such as multipath fading and environmental interference. However, we would like to clarify that these effects fall outside the scope of our current study. Our work is not intended to design or evaluate a new communication system; rather, it focuses on improving the machine learning based prediction of DC bias in DCO-OFDM transmission. For this purpose, we employed the dataset provided by Purnita et al. [10], which has already been validated and published in the literature. We acknowledge that extending the evaluation to more complex channel conditions is an interesting direction for future research. We mentioned this concern in the revised manuscript in Data description subsection. Please see the highlighted PDF on page 8, line 269:

The simulation model used to generate the dataset considered only an AWGN channel. This choice is justified because AWGN is the dominant noise source in indoor optical wireless communication environments, and the primary objective of DC bias optimization is to mitigate clipping noise at the transmitter before signal propagation. The BER benchmark was therefore selected to ensure clipping noise remained confined to an acceptable level.

Response to Reviewer #3

Comment 1: Impact on data size

The authors have given acceptable reasons and advised that the impact of data size on model performance would be conducted as further research. There are some established methods noted by the authors (For example, a statistical measure of reliability such as Kappa index). Adequate citations on this could be included as part of the conclusion section. This was not reflected appropriately in the present manuscript.

Response:

Thank you for your observation. In the previous revision we have reasoned that in the response to reviewer comments only. In this round of revision, we ensured tha

---

## [Decision Letter · Decision Letter 2]

22 Oct 2025

A Robust Machine Learning Approach for DC Bias Prediction in DCO-OFDM Based Li-Fi Systems

PONE-D-24-52879R2

Dear Dr. Salman,

We’re pleased to inform you that your manuscript has been judged scientifically suitable for publication and will be formally accepted for publication once it meets all outstanding technical requirements.

Kind regards,

Miguel A. Fernández, Ph.D.

Academic Editor

PLOS ONE

Additional Editor Comments (optional):

Reviewers' comments:

Reviewer's Responses to Questions

**Comments to the Author**

1. If the authors have adequately addressed your comments raised in a previous round of review and you feel that this manuscript is now acceptable for publication, you may indicate that here to bypass the “Comments to the Author” section, enter your conflict of interest statement in the “Confidential to Editor” section, and submit your "Accept" recommendation.

Reviewer #4: All comments have been addressed

2. Is the manuscript technically sound, and do the data support the conclusions?

Reviewer #4: Yes

3. Has the statistical analysis been performed appropriately and rigorously? 

Reviewer #4: Yes

4. Have the authors made all data underlying the findings in their manuscript fully available?

Reviewer #4: Yes

5. Is the manuscript presented in an intelligible fashion and written in standard English?

Reviewer #4: Yes

6. Review Comments to the Author

Reviewer #4: (No Response)

7. PLOS authors have the option to publish the peer review history of their article (what does this mean?). If published, this will include your full peer review and any attached files.

Reviewer #4: No

---

## [Editor Report · Acceptance letter]

PONE-D-24-52879R2

PLOS ONE

Dear Dr. Salman,

I'm pleased to inform you that your manuscript has been deemed suitable for publication in PLOS ONE. Congratulations! Your manuscript is now being handed over to our production team.

Kind regards,

on behalf of

Dr Miguel A. Fernández

Academic Editor

PLOS ONE